# Maturing pharmacogenomic factors deliver improvements and cost efficiencies

Joseph P. Jarvis[1], Scott E. Megill[1], Peter Silvester[2] and Jeffrey A. Shaman[1]

[1]Coriell Life Sciences, Philadelphia, PA, USA and [2]Thermo Fisher Scientific, Waltham, MA, USA

pharmacogenetics; medication management; personalized medicine; medication safety; adverse drug reactions; polypharmacy

**Corresponding author:**
Jeffrey A. Shaman,
E-mail: jshaman@coriell.com

## Abstract

An ever-expanding annotation of the human genome sequence continues to promise a new era of precision medicine. Advances in knowledge management and the ability to leverage genetic information to make clinically relevant, predictive, diagnostic, and targeted therapeutic choices offer the ability to improve patient outcomes and reduce the overall cost of healthcare. However, numerous barriers have resulted in a modest start to the clinical use of genetics at scale. Examples of successful deployments include oncologic disease treatment with targeted prescribing; however, even in these cases, genome-informed decision-making has yet to achieve standard of care in most major healthcare systems. In the last two decades, advances in genetic testing, therapeutic coverage, and clinical decision support have resulted in early-stage adoption of pharmacogenomics – the use of genetic information to routinely determine the safety and efficacy profile of specific medications for individuals. Here, through their complicated histories, we review the current state of pharmacogenomic testing technologies, the information tools that can unlock clinical utility, and value-driving implementation strategies that represent the future of pharmacogenomics-enabled healthcare decision-making. We conclude with real-world economic and clinical outcomes from a full-scale deployment and ultimately provide insight into potential tipping points for global adoption, including recent lessons from the rapid scale-up of high-volume test delivery during the global SARS-CoV2 epidemic.

## Impact Statement

The scalable, broad utilization of genetic testing in personalized medicine requires many factors working together to achieve value for patients, providers, and payors, and to avoid disruption of existing clinical workflows. We catalogue those factors and describe the historic context of each towards maturation and scalable deployment. Specifically, we make clear the compelling case that the tipping point has been reached for the five factors – clinical utility, laboratory technology, user acceptance, implementation models, and economic value – in favor of large scale pharmacogenomic testing in a global context. As a treatise on the current state of clinical pharmacogenomics, health systems, payors, and population caretakers can leverage this research as guiding evidence pointing squarely in favor of acting now to stem the tide of rising healthcare costs. Codified in a key central figure, this paper promises to be a seminal reference to advance this field of genetic science towards becoming an international clinical standard of care. The use of genetics as standard of care has the potential to provide all stakeholders a low-cost solution to poor health and rising healthcare costs.

## Introduction

Successful integration of precision medicine into standard care requires structured approaches and maturity across the biological, technological, regulatory, educational, and clinical landscapes. It also requires acceptance and support from stakeholders across all aspects of healthcare delivery. Pharmacogenomics (PGx), defined as the subset of precision medicine dealing with the safety, efficacy, and interactions of established pharmaceutical interventions as informed by genetics (Garrod, 1908; Vogel, 1959; Roden et al., 2011, 2019), has undergone significant maturation. Notably, over roughly the past two decades, each of five key features in the field of PGx has attained sufficient levels of maturity and value to reach a genuine tipping point for the large-scale adoption and implementation of genome-informed clinical decision-making. The goal of this review is to describe the historical progression, intersecting trajectories, and current readiness of critical features of PGx implementation by documenting impactful advancements, and ultimately the establishment of the economic value of broad panel, population-scale PGx testing.

Specifically, we identify and track five factors: clinical utility, laboratory technology, user acceptance, implementation models, and economic value as requirements for viable and valuable deployment of PGx-enabled precision medicine (Figure 1). In particular, we define the following:

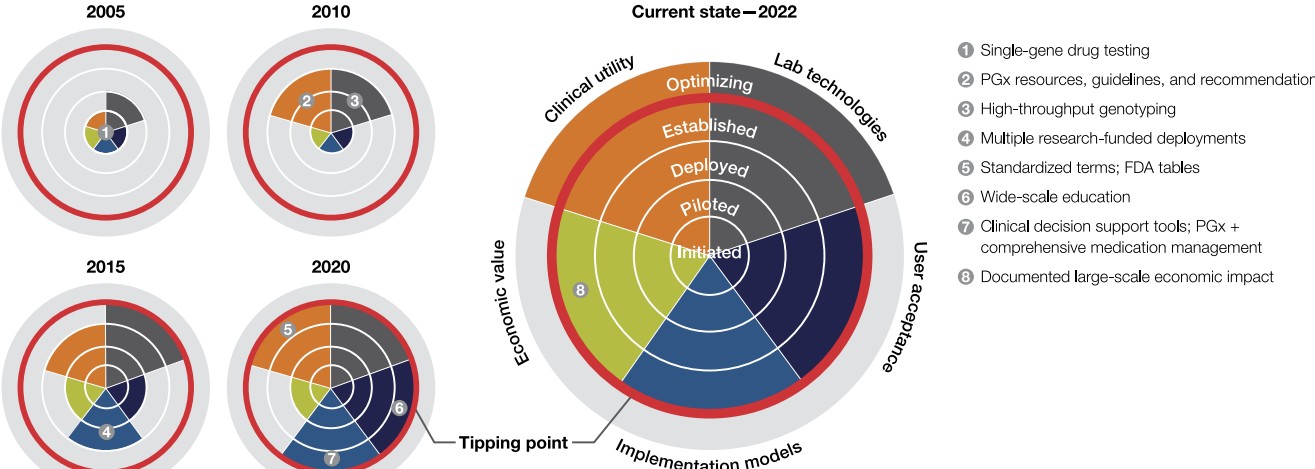

**Figure 1.** Maturation and readiness of five factors driving large-scale PGx implementation. Graphical depiction of the maturation of five key factors: clinical utility (orange), laboratory technology (gray), user acceptance (dark blue), implementation models (light blue), and economic value (green), progressing through the five specific stages of maturation: initiated, piloted, deployed, established, and optimizing, during five critical time periods: 2005, 2010, 2015, 2020, and current state – 2022. Listed at right and plotted as numbered bullets, are significant developments that drove the progression of each factor through the maturation stages. The tipping point (dark gray ring) denotes where sufficient maturation of each factor supports large-scale PGx implementation. Plot 2005 represents the initial state of maturity up to the year 2005, with lab technologies representing the most advanced of the five factors. The 2010 chart captures the growth since 2005, notably in clinical utility – the development of PGx resources, guidelines, and recommendations based on research on PGx health outcomes. In the next time period ending in 2015, multiple research implementation models have been deployed and laboratory technologies hit the tipping point. By 2020, significant developments in clinical utility, user acceptance, and implantation models, drove these factors to the tipping point, as well. By 2022 with the publication of the first large-scale economic impact publication, each factor had reached readiness for large-scale PGx implementation.

- *Clinical utility:* The ability of meaningful clinical interpretations, recommendations, and guidelines based on existing research and clinical understanding, to positively impact health outcomes (Khoury, 2003).
- *Laboratory technology:* encompasses genotyping and sequencing technologies, and the standardization, regulation, and efficient production of laboratory data, and rapid delivery of results at scale.
- *User acceptance:* the degree to which patients, healthcare providers, and payors understand, embrace, and engage with genomic testing and the resulting insights.
- *Implementation models:* the methods, processes, and practices employed to utilize pharmacogenomics within health delivery systems.
- *Economic value:* the demonstrated real-world benefits of cost savings and patient health improvement directly attributable to the application of PGx-enabled precision medicine.

## Global readiness for large-scale PGx implementation

### 2005: Clinical utility drives PGx innovation

In the immediate years before and after the completion of the human genome reference sequence (Venter et al., 2001; Nurk et al., 2022), most precision medicine advancements were confined to the world of research (Weinshilboum and Wang, 2004; Kalow, 2005; Somogy, 2008; Offit, 2011). For PGx, these efforts were often driven by catastrophic clinical outcomes or other obvious effects of treatment using relatively small groups of patients (Steiner et al., 2005; Ciccolini et al., 2006; Maitland et al., 2006). They also centered primarily on the dosing recommendations from pharmacokinetic studies (Kirchheiner et al., 2001; Gardiner and Begg, 2005). Largely implemented in the context of academic medical centers and executed by global experts in specific disease fields, there was relatively little standardization in DNA testing or regulatory

oversight and even less integration with the wider care delivery apparatus (McKinnon et al., 2007; Swen et al., 2007; Shuldiner et al., 2013). However, the very real successes that emerged helped to build the case for the clinical benefits of adding genomic information to the development of personalized treatment plans (Streetman, 2007).

From a clinical utility perspective, early enthusiasm for PGx as a viable approach to precision medicine was fed by these successes. For example, the regulatory approval of certain medications with molecularly driven patient-selection strategies (e.g., Herceptin) (Roukos, 2011; Sawyers, 2019) and the preemptive identification of sensitivities to medications (e.g., Mercaptopurine, Debrisoquine) (Mahgoub et al., 1977; Weinshilboum and Sladek, 1980) were published during this period. Such findings created a nascent understanding of the potential economic value of more specific strategies for prescribing as well. Furthermore, progress in the identification of molecular pathways associated with pharmacokinetics and pharmacodynamics of existing drugs and the realization that the genetic component of PGx represented a tractable set of research questions, encouraged both a diversity of researchers and early commercial entities to continue exploration in the space (Zhang and Nebert, 2017). Especially important from a statistical perspective was that drug metabolism seemed to involve as few as eight genes (Fernandez et al., 2012; Tremaine et al., 2015), some of which were fairly well understood. This, in turn, drove the realization that many targets of pharmaceutical interventions could involve genome-informed approaches from the very first stages of screening, thus greatly facilitating the interpretation of sequence information. This also meant that feasible sample sizes of patients using targeted assays were likely to produce novel, or at least confirmatory, evidence of medical relevance and clinical actionability.

As the field picked its low-hanging fruit, *a priori* information emerging from molecular studies allowed targeted interrogation of additional genetic variants of interest and their relationship to the

metabolism of additional compounds. Thus, the clear variation in liver metabolism due to inherited genetic factors supported full-scale research programs exploring the effects of novel variation in specific genes on clinical outcomes (Alexanderson et al., 1969; Sconce et al., 2005; Ahmed et al., 2016; Yang et al., 2017). Subsequent systematic investigations ultimately uncovered clinical reasons for deaths associated with capecitabine (Syn et al., 2016), 5-fluorouracil (5-FU), and other fluoropyrimidines (Soong and Diasio, 2005), as well as PGx factors involved in the rare but devastating Stevens–Johnson syndrome (SJS) (Sukasem et al., 2018).

Importantly, early work established the fundamental unit of PGx interpretation as the drug–gene or, more specifically, drug–variant pair. This insight was key since altered enzyme function must be placed in the context of the pharmacodynamic and pharmacokinetic properties of specific compounds. That is, each functional genomic variant has the potential to impact the processing of multiple compounds in different ways. What is 'slow' for one process may be entirely sufficient for the degradation pathway (i.e., catabolism) of another. Thus, standardized interpretations based on metabolizer status are required for a scaled approach to testing but require substantial finesse in the delivery of interpretations as well as a significant educational component for providers. It also implies that a huge amount of expertise, presented in databases such as PharmGKB and guidelines developed by the Clinical Pharmacogenetics Implementation Consortium (CPIC) and the Dutch Pharmacogenomics Working Group (DPWG), will continue to be needed in order to curate the primary research literature into something useful for clinicians.

One way that these subtleties were addressed, and standardization began to develop, was the emergence and proliferation of drug labels that included genetic information (Frueh et al., 2008; Kim et al., 2021), and perspectives and guidance on PGx-guided drug development (Lesko and Woodcock, 2002; FDA, 2005). Since drug labels are produced by the drug maker but circulated under the approval of regulatory agencies (e.g., US FDA), the beginnings of regulatory oversight began to take shape even if only in a rudimentary way. This was not without controversy as other models of oversight showed promise as well, including Informed Cohort Oversight Boards (ICOB) (Keller et al., 2010; Kohane and Taylor, 2010; Holm and Taylor, 2012), Pharmacogenomics Advisory Groups (PAG) (Gharani et al., 2013), and other multi-disciplinary bodies such as CPIC and the DPWG (Abdullah-Koolmees et al., 2020; Yoon et al., 2020; Pritchard et al., 2022). However, a balance in labor intensity and efficiency given realistic logistical constraints tended to steer system-wide developments along paths of least resistance. One unintended consequence was that, since standard practice for drug label content was to draw heavily from previous labels, whole informational sections on existing drug labels were often simply cut and pasted from an original submission into guidance for structurally similar biochemical compounds (e.g., tricyclic antidepressants). While expedient, this practice ultimately obfuscated the drug–variant relationship for some families of drugs but also created a certain level of credibility for genome-informed decision-making around prescribing.

In parallel to these clinically and regulatory-driven advancements, technologically driven standardization of genome-wide marker arrays, data management, and approaches to sequencing technologies also proceeded. New techniques ultimately replaced previously standard laboratory practice for assaying variants that relied on inefficient, manual processes of DNA extraction, Sanger sequencing, variant determination, and reporting of results in ways that were simply not scalable (Tipu and Shabbir, 2015; Heather and Chain, 2016). In practice, however, technical scalability emerged quickly where free market incentives drove investment in innovation and consolidation, especially vis-à-vis the replacement of output formats such as chromatograms which required manual review by expert technicians in order to extract meaningful assay results and inhibited automated report generation.

As a result of increased automation and scale-friendly approaches to PGx reporting, life sciences companies, the pharmaceutical industry, private laboratories, and payors responded by further streamlining other processes. For example, DNA sequencing and genotyping companies were selling instruments and assays to a wide variety of customers including as part of Direct-to-Consumer (DTC) products (Allyse et al., 2018). This, in turn, fed user adoption and also produced a variety of strategies for implementation addressing many logistical challenges and generated initial discussions of cost-effectiveness (Phillips and Van Bebber, 2004).

## 2010: Scalable laboratory technologies drive PGx maturity

By 2010, with the research community continuing to rapidly expand the primary literature in PGx, and to standardize interpretations, additional players were drawn to the emerging commercial market of PGx testing. Given the relative lack of regulatory oversight, generally high (though inconsistent) reimbursement rates, and emerging efficiencies in report development, multiple models of delivering genetic results and their PGx interpretations were attempted in the commercial markets (Hresko and Haga, 2012; Frueh, 2013). These varied in their sophistication and the degree of involvement of researchers, clinicians, providers, and the use of various publicly available tools and resources. This broad experimentation drove increasing diversity in the specifics of interpretation, substance, and wording (Wright et al., 2011; Roberts and Ostergren, 2013) and so ultimately underscored the need for additional standardization and oversight of key processes and best practices.

Between 2005 and 2010, in addition to content derived from drug labels, other sources of authoritative annotations such as 'Flockhart Tables' (Flockhart and Oesterheld, 2000; Indiana University, 2021), PharmGKB (Sangkuhl et al., 2008), results from the DPWG (Swen et al., 2008), CPIC guidelines (Relling and Klein, 2011), and the output from several large-scale research projects such as the Coriell Personalized Medicine Collaborative (CPMC) (Keller et al., 2010) also emerged as trusted sources of clinical interpretation. As a result, the consensus around common nomenclature for metabolizer phenotypes, clinical language, and guidance became necessary and was deployed (Kalman et al., 2016; Caudle et al., 2020), thus knitting together (and so maturing) multiple resources necessary for scaled adoption of PGx interpretation.

Importantly, the dynamic nature of PGx research resulted in products that did not keep up with the subtle advances in multiple domains and the surrounding scientific consensus. These soon became obsolete without continued review and refinement. This, in turn, attracted US regulatory scrutiny and demonstrated the complex requirements around the education of physicians (Stanek et al., 2012; Rohrer Vitek et al., 2017, 2021; Karas Kuzelicki et al., 2019), pharmacists (McCullough et al., 2011; Benzeroual et al., 2012; Nickola et al., 2012), nurses (Calzone et al., 2010), and genetic counselors (Haga et al., 2012; Loudon et al., 2021) that

would ultimately lead to successful, scaled implementation. In the end, meaningful training and education for a variety of healthcare providers were established as critical to the health and economic benefits of emerging delivery models (Green and Guyer, 2011).

Ongoing technological advances led to increasingly scalable, off-the-shelf assays for genetic variations in genes encoding drug metabolizing enzymes and transporters (ADME) or in genes encoding drug receptors (Daly et al., 2007; Deeken, 2009; Burmester et al., 2010; Fernandez et al., 2012; Arbitrio et al., 2018; Agapito et al., 2020). These high-throughput technologies included real-time PCR (e.g., TaqMan OpenArray PGx Express Panel, Thermo Fisher Scientific, Waltham, MA), microarrays (e.g., Affymetrix DMET Plus Array, Thermo Fisher Scientific; VeraCode ADME Core Panel, Illumina, San Diego, CA), and mass spectroscopy arrays (e.g., iPLEX ADME PGx Panel, Sequenom, San Diego, CA). Additionally, some degree of standardization of the output of genotyping technologies was also established (e.g., variant call format, VCF) in addition to the continued divergence of cost and base pair production speed often compared to Moore's law (Wetterstrand, 2021). Firm file formats for various chemistries, specialized hardware to facilitate the speed and scalability of certain assays, and quality scoring standards were also introduced. However, while this period of advancement laid the groundwork for further scaled automation and allowed for the application of yet more stringent laboratory process requirements to be satisfied (i.e., CAP/CLIA), very little progress was made in integrating this information with existing electronic health record infrastructures and the wider clinical care apparatus.

### 2015: Implementation models emerge from evidence-derived best practices

With the emergence of best practices now underway across all five of the major focus areas, a wide variety of viable implementation models, research studies, and commercial products began gaining traction between 2010 and 2015 (Daly, 2012; Dunnenberger et al., 2015; Kaufman et al., 2015; Rogers et al., 2020). Some offered products geared toward specific patient groups and others were built around combinations of stacked CPT codes without particular emphasis on narrow disease indications. These efforts were particularly aided by an ever-increasing volume of primary literature, which continued to drive improvements in the clinical utility of results. While the pipeline of publications would ultimately need several more years to reach broad levels of applicability for implementation at the population level, other areas such as laboratory technology reached sufficient maturity to be considered fully established and scalable during this period. In particular, various genotyping chemistries achieved industrial levels of efficiency. These included various versions of high-throughput array technology as well as several PCR-based approaches and whole genome sequencing.

One particularly notable development in implementation models was a substantial increase in regulatory oversight for DTC products. Following a rapid escalation in consumer advertising in 2013, 23andMe, the largest DTC provider of genomic testing, received warnings from the FDA about providing medical advice directly to its customers without premarket regulatory participation (Annas and Elias, 2014; Delaney and Christman, 2016). The company complied with the request and received approval to begin providing limited results to its customers via its dynamic, password-protected portal (US Food and Drug Administration, 2018; 23andMe, 2019).

**Table 1.** Examples of research-funded deployments exploring implementation models

| | |
|---|---|
| CPMC | Coriell Personalized Medicine Collaborative (2007) |
| PREDICT | Pharmacogenomic Resource for Enhanced Decisions in Care and Treatment (2010) |
| PGRN | Pharmacogenomics Research Network, Translational Pharmacogenetics Program (2011) |
| PG4KDS | St. Jude Children's Research Hospital's Clinical Implementation of Pharmacogenetics (2011) |
| | University of Chicago, The 1,200 Patients Project (2011) |
| | Cleveland Clinic, Personalized Medication Program (2011) |
| ICAPS | The International Consortium for Antihypertensives Pharmacogenomics Studies (2012) |
| RIGHT | Right Drug, Right Dose, Right Time – Using Genomic Data to Individualize Treatment (2012) |
| eMERGE-PGx | Electronic Medical Records and Genomics Network – Pharmacogenomics Project (2013) |
| IGNITE | Implementing GeNomics In pracTicE (2013) |
| INGENIOUS | INdiana GENomics Implementation: an Opportunity for the Under Served (2014) |
| PreCISE-Rx | Pharmacogenomics-guided Care to Improve the Safety and Effectiveness of Medications (2016) |
| ACCOuNT | African American Cardiovascular pharmacogenetics CONsorTium (2016) |

Other companies such as Translational Software, OneOme, and Coriell Life Sciences developed a reporting approach based on more traditional laboratory services that leveraged existing oversight mechanisms. Laboratories operating under the regulatory guidance of CLIA could engage with these third-party reporting services to provide interpreted PGx panels to their customers. Critically, this differed from the DTC approach by delivering results to an ordering clinician rather than to a patient directly. This assured that PGx guidance was received by a physician as clinical decision support within the practice of medicine – an area not typically regulated by the FDA.

Large-scale research projects also continued to innovate and inform the primary literature on subjects as varied as the potential economic impacts of PGx testing, the ethics of delivering genomic results, and levels of action taken by participants upon learning their results (Diseati et al., 2015; Scheinfeldt et al., 2016; Kusic et al., 2020). Such projects included IGNITE (Weitzel et al., 2016), Vanderbilt's PREDICT, which published an overview of the PGx environment in the United States (Pulley et al., 2012; Volpi et al., 2018), and other implementation models (Table 1). The CPMC, through its partnership with the US Air Force, also continued to both expand its portfolio of reports and explore novel methods of result delivery and healthcare provider engagement (Delaney et al., 2017). Of note was their use of a nationwide network of trained genetic counselors with appropriate state-level credentials to offer on-demand review of any reported results via telemedicine at a participant's request.

Interest and trust began to grow in genome-informed approaches to care and especially for PGx results which have biological and biochemical explanations. However, clinical uptake continued to be limited by challenges in test coverage and reimbursement (Rogers et al., 2020). Related companies and products, such as 23andMe and AncestryDNA, also contributed to a greater

public awareness and familiarity with DNA analysis and the practical utility of the genome, but also to public concerns around providing samples and the use of the resulting information. This, along with physician education initiatives such as Manchester University's pharmacogenomics master's degree program (Manchester University, 2016), contributed to wider user acceptance and provider confidence.

The final critical development during this period was the first 'major' publication supporting broad-scale economic value (Brixner et al., 2016). While some highly focused products had previously demonstrated value for small groups with specific diagnoses or therapies like antidepressant or antipsychotic medications (Altar et al., 2015; Winner et al., 2015), warfarin treatment (Mitropoulou et al., 2015), or use of fluoropyrimidines (Deenen et al., 2016), Brixner et al. began to address how a study could be constructed to explore the economic benefits that would be expected to accrue given large-scale implementation of PGx reporting. Other groups contributed to this speculation as well, highlighting the roles of public policy and national healthcare systems in the emerging ecosystem of PGx interventions and their cost/benefit dynamics (Mitropoulou et al., 2020).

## 2020: Broader adoption of best practices and transition to standard of care

By 2020, with the majority of the technical challenges of genetic assay design and development established, the stage was set for a period of consolidation, coordination, and distillation of the advances required for scaled implementation (Lam and Scott, 2018). For example, clinical utility reached sufficient maturity to support population-level products, an especially important milestone being the publication of updated FDA tables (U.S. FDA, 2021, 2022). This period also saw clear guidelines and standardized terminology arise as a consensus from CPIC's long-running work in those areas (Caudle et al., 2017, 2020). While continued dialogue on such language is likely to take place, the establishment of an authoritative source for suggested report language was nonetheless a major advancement. User acceptance – from both healthcare providers and patients – also reached a tipping point with the implementation of wide-scale educational initiatives (Karas Kuželički et al., 2019), the release of draft local coverage determinations, and through a critical mass of testing that fostered increasing levels of healthcare provider trust.

Finally, implementation models too reached the level of efficiency and refinement needed for scaled delivery. In particular, the application of a clinical decision support system (CDSS) in a clear comprehensive medication management (CMM) framework became firmly established as multiple entities applied this combined approach (Elliott et al., 2017; Jones et al., 2018; Jarvis et al., 2022). Thus, as PGx delivery strategies reached sufficient maturity, evidence supporting value-based arguments for broad-scale implementation similarly arose. What began as a high-value proposition to avoid rare but catastrophic outcomes in specific groups of patients had evolved into a much broader intervention involving more moderate effect sizes applicable to large populations and varied healthcare needs.

One implementation, in particular, was begun during this period which, though not a truly randomized controlled trial, represented the first full evaluation of the effects of PGx on an entire healthcare system (Jarvis et al., 2022). By combining a CDSS and PGx + CMM, and involving additional healthcare providers in the form of specially trained pharmacists, it was able to directly demonstrate a broad reduction of errors in the clinic, the optimization of therapy plans, and overall avoidance of severe and predictable medical outcomes in the patient population. As such it, and other work currently underway, are addressing the final remaining gap: establishing the economic value of personalized prescribing at scale. What has already been established, however, are multiple attendant benefits to both patients and healthcare systems through the deployment of composite approaches in the context of CMM. Specifically, the authors observed significant savings for integrated healthcare systems as well as adjustments in patient healthcare resource utilization when a genome-informed CDSS was put in place. It demonstrated that, in an older population of adults, and in the context of a large degree of polypharmacy, per patient per month costs were reduced by an average of 218 USD. At the level of the entire population, the reductions totaled 37 million USD over 32 months.

With the establishment of real-world value, PGx successfully moved out of the exclusive world of academic medical institutions and into an emerging academic-industry ecosystem with great promise for improving multiple levels of the healthcare system. While logistical hurdles inherent in the unique structure of the US healthcare market remain challenging, certain 'vertically integrated' healthcare models and growing interest in nationalized healthcare systems in Europe and the Middle East have the opportunity to lead the way in providing models for optimized PGx delivery. What these systems share is a vested interest in long-term outcomes and data-driven, value-based care.

## 2022: Tipping point reached for large-scale PGx implementation

It has been clear since the early stages of large-scale research projects (e.g., the CPMC) that physicians have neither the time nor the incentive to track the moving target of relationships between the genome and medication outcomes. Thus, some degree of specialized training is required to carefully curate the literature, cultivate an appropriate reaction to published results, and present it to physicians for their consideration. This likely requires a combination of privately funded, market-driven approaches that engage true curation experts as well as government-funded and publicly available resources. It remains to be seen how such a system will emerge and evolve but all of the pieces of the puzzle have achieved levels of maturity that should drive further innovation with the appropriate economic stimuli.

In addition, the critical developments required to unlock demonstrated value revolve around the scalability of testing and interpretation and the delivery of high-quality information to the appropriate healthcare provider across the continuum of care. In some cases, this may be a specialist; in others, a primary care physician; in still others, the engagement of well-trained but under-utilized resources such as pharmacists. The general principle that has emerged as a best practice is to provide actionable information to those with the time and training to suggest a treatment course in consultation with all those involved in care delivery. The challenges are largely logistical in just how exactly to accomplish these goals in a fragmented healthcare delivery apparatus.

Additional opportunities to refine and optimize the personalized nature of the emerging best-practice PGx delivery model should also be noted. For example, variation in key allele frequencies across population samples – including those which have not been comprehensively studied – may require tailoring, curation, and reporting for certain geographic regions. Some indication of this potential reality is seen for populations including substantial

diversity. These include those in US states like Hawaii, which have a large representation of individuals of Asian descent, the differences in *CYP2C19* alleles suggest a much larger potential impact of poor metabolism of corresponding drugs (Bellon and Raymond, 2021; Alrajeh and Roman, 2022). As such, public health officials and other regulatory agencies as well as physicians working with such patient populations may need regional strategies to ensure proper messaging around the magnitude of potential effects. This phenomenon is also likely to shift value propositions for healthcare systems serving these groups. As some individuals may disproportionately benefit from genomic testing, some systems may see disproportionate savings as well.

Finally, the recent massive global investment in point-of-care testing and results delivery necessitated by the 2020 COVID-19 pandemic also presents a unique opportunity to observe the power and value of rapid and scalable testing at the population level (Mercer and Salit, 2021). Re-purposing the infrastructure centered around viral testing to address other pressing public health concerns represents a major opportunity that should not be overlooked. Molecular diagnostic platform manufacturers and laboratories demonstrated their real-time ability to answer the call by quickly increasing the volumes and speeds at which samples could be collected, coordinated, and processed by increasing capacity and adopting new, more efficient testing procedures (Kriegova et al., 2020; La Marca et al., 2020; Sanyaolu et al., 2020; Zhang et al., 2020; Johns Hopkins Center for Health Security, 2022). Tapping this latent potential in other areas, securing adequate funding, and publicizing the existing documentation of critical returns-on-investment are all necessary to demonstrate the future of all sorts of testing strategies, including those related to PGx.

## Genome-informed medication management is the future of modern effective healthcare

Ultimately, the clear improvements in patient care and cost savings for the healthcare system produced by thoughtful, comprehensive PGx + CMM strategies are compelling as a new standard of care. Streamlining logistics, physician educational initiatives, and the engagement of other healthcare provider groups (e.g., pharmacists) will help to reduce the cost of PGx as an intervention. Thus, when solutions to the management of genomic data and interpretations in the context of existing electronic health records systems can be achieved, a full and working ecosystem empowering physicians and patients in the management of their medications will be complete.

Such a system will always require expert curation. Moreover, geographical variation in allele frequencies is likely to drive regional product sets that best capture local genomic diversity. These are also likely to feed additional research programs identifying new variants of consequence, as well as facilitate the development of highly efficient testing strategies that will reduce costs and increase precision. It remains to be seen what scientific and regulatory bodies will emerge as authoritative.

Having hit the tipping point where PGx is proven to contribute to meaningful improvements in healthcare delivery, the stage is set for widespread adoption and optimization of its use as a standard of care. Additional implementations will add value and utility; clinical decision support tools will also continue to improve.

The rising and unsustainable economic burden of healthcare provision is matched only by raising public awareness, expectations, and access to targeted, effective, personalized care. Genetically informed prescribing and active medication management

offers one of the few realistic mechanisms to effectively reduce overall healthcare system costs while also improving individual patient care and outcomes. Historic barriers to scaled deployment of pharmacogenomics combined with CMM have been largely removed as outlined in this review. Furthermore, PCR testing infrastructure built to support a scaled response to the COVID pandemic, combined with an increasing public awareness of the value of genome-informed decision-making, is now both available and suitable to be re-tasked toward population-scale deployment of PGx. The time is now for healthcare systems, employers, and governments to partner and establish genome-informed medication management as a new standard of care.

**Open peer review.** To view the open peer review materials for this article, please visit http://doi.org/10.1017/pcm.2022.3.

**Data availability statement.** Data availability is not applicable to this article as no new data were created or analyzed in this study.

**Acknowledgements.** We graciously acknowledge and thank Tricia Kenny for her thoughtful and critical review of various drafts of the manuscript. We also thank Dr. Arul Prakasam Peter for identifying additional citations.

**Author contributions.** J.P.J., S.E.M., P.S., and J.A.S. participated in the conception, writing, and proofing of the manuscript. All authors have read and agreed to the published version of the manuscript.

**Financial support.** This research received no specific grant from any funding agency, commercial, or not-for-profit sectors.

**Competing interests.** P.S. is Senior Vice President and President (retired), Life Sciences Solutions, Thermo Fisher Scientific, and has stock ownership in Thermo Fisher Scientific. J.P.J. receives compensation from and has an equity interest in Coriell Life Sciences. S.E.M. and J.A.S. are employed by, receive a salary from, and have an equity interest in Coriell Life Sciences.

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
