## [Reviewer Report]

*Comments to Author*: Please see the word document.

Review of “Maturing pharmacogenomic factors deliver improvements & cost efficiencies”

The authors provide a review the current state of pharmacogenomic testing technologies, information tools and implementation strategies for pharmacogenomics enabled healthcare decision-making. In addition, they discuss outcomes from a full-scale deployment and project future trends.

Summary: the authors provide an interesting historical overview which includes a significant number of references. However, the authors may want to consider extracting specific cases from the references to illustrate and clearly support broad statements particularly in the early sections of the paper. In the latter sections e.g. Implementation Models the examples cited are helpful for readers. In its current form, the submission seems to have many characteristics of a ‘white paper.’

Line 38 – The authors provide their own definitions for clinical utility and four additional factors. Consider alternate definitions from the CDC or others which could be referenced. Are these definitions preferred by their organizations, Coriell Life Sciences and/or Thermofisher?

Line 41 – Utility is defined here by the authors as “positive.” Have the authors considered “negative” utility? i.e. false positive, false negatives, and other harms? Usefulness or value may be more appropriate? Why not use a prior published definition rather than a new variation?

Line 45 – User Acceptance, nor the other 4 factors, mention payer acceptance. This would seem to be an important factor

Line 54 – The authors state, “In the immediate years before and after the completion of the human genome reference sequence precision medicine advancements were confined to the world of research.” The genome project completed in 2003, and the first FDA PGx for patient use was in 2005. Recommend revising this section - optimally presenting data - or a reference to support this specific point.

Line 68 - the rescue “of” [?] certain medications with genetic implications (e.g., Herceptin) (Roukos 2011). Is this a grammatical typo or do the authors mean “by”?

Line 118 - Please provide a references(s), or an illustration(s), for “One unintended consequence was that, since standard practice for drug label content was to draw heavily from previous labels, whole informational sections on existing drug labels were often simply cut and pasted from an original submission into guidance for structurally similar biochemical compounds.”

Line 130 – “In practice however, technical scalability emerged quickly where free market incentives drove innovation and consolidation.” Please cite a “free market incentive” illustration or reference for this point.

Line 136 – Given the emphasis on direct-to-consumers in this paragraph please provide an illustration or a reference specific to the point that “payors responded by further streamlining other processes” and support the comment on cost-effectiveness discussions.

Line 172 – Evidence or data to support the statement, “Interestingly, this additional regulatory scrutiny had a cooling effect and caused a retreat for DTC PGx offerings.”

Line 234 – “However, clinical uptake continued to be limited by challenges in test coverage and reimbursement.” The Rogers publication is cited which is fine. However, supporting detail, a case illustration, etc. would be helpful for the reader.

Line 292 – Please specify more clearly the monetary impact “hundreds of dollars could be saved per patient per month. At the level of the entire population, the savings totaled in the millions of USD.”

Line 838 – typo. Repeat “of the of the”

---

## [Editor Report]

*Comments to Author*: This is an excellent and comprehensive overview on various challenges of PGx implementation. In addition to reviewer 2 and as the heading of this manuscript includes "cost efficiencies", some more emphasis should be given on the role of payers and implementation of PGx into a reimbursement system dependent on drug labelling (e.g. mandatory vs. recommended genetic testing).